# Training diversity promotes absolute-value-guided choice

**Levi Solomyak**[iD]*, **Paul B. Sharp, Eran Eldar**[iD]

The Hebrew University of Jerusalem, Jerusalem, Israel

* Levi.solomyak@mail.huji.ac.il

**Data Availability Statement:** All human data is freely available as a Dryad, Dataset, https://doi.org/10.5061/dryad.1rn8pk0xr. All code written in support of this publication is publicly available at https://github.com/lsolomyak/training_diversity_promotes_absolute_value_guided_choice

## Abstract

Many decision-making studies have demonstrated that humans learn either expected values or relative preferences among choice options, yet little is known about what environmental conditions promote one strategy over the other. Here, we test the novel hypothesis that humans adapt the degree to which they form absolute values to the diversity of the learning environment. Since absolute values generalize better to new sets of options, we predicted that the more options a person learns about the more likely they would be to form absolute values. To test this, we designed a multi-day learning experiment comprising twenty learning sessions in which subjects chose among pairs of images each associated with a different probability of reward. We assessed the degree to which subjects formed absolute values and relative preferences by asking them to choose between images they learned about in separate sessions. We found that *concurrently* learning about more images within a session enhanced absolute-value, and suppressed relative-preference, learning. Conversely, *cumulatively* pitting each image against a larger number of other images across multiple sessions did not impact the form of learning. These results show that the way humans encode preferences is adapted to the diversity of experiences offered by the immediate learning context.

## Author summary

Learning relative preferences between a pair of options is effective in guiding choice between them, but might lead to error in choosing between options that have not been paired against each other even if we know each option well. This problem of generalizing relative preferences to novel decision contexts increases as the number of options gets larger, since the more options there are the more likely we are to encounter choices between new sets of options. To solve this problem, people may learn the expected reward associated with each individual option—that is, its 'absolute value', by means of which any pair of options can be compared. Thus, we hypothesized that the more options a person learns about, the more likely they would be to form absolute values as opposed to relative preferences. We constructed a novel multi-day reward learning experiment to specifically test this hypothesis. We found that concurrently learning about more images indeed enhances absolute-value learning and suppresses relative-preference learning. The

**Funding:** This work has been made possible by NIH grants R01MH124092 and R01MH125564 (to E.E), an ISF grant 1094/20 (to E.E), and the US-Israel BSF grant 2019802 (to E.E.). The funders had no role in study design, data collection and analysis, decision to publish, or preparation of the manuscript.

**Competing interests:** The authors have declared that no competing interests exist.

findings clarify what learning conditions promote the formation of generalizable preferences that can help reach optimal decisions across different contexts, an ability that is vital in the real world where experience is limited and fragmented across multiple continuously shifting contexts.

## Introduction

A large body of decision-making research suggests that humans learn from experience the expected value of different available options (hitherto referred to as these options' absolute values; [1–5]). That is, as people try out different options and observe their reward outcomes, it is thought that they track the average reward associated with each option, and this allows them to choose options with higher expected value. The main benefit of this so-called *value learning* is that it makes it easy to choose from any set of options, including options that have not been previously considered in relation to one another, simply by comparing their absolute values. In this sense, absolute values act as a "common currency" that serves to generalize preferences across contexts that offer different combinations of options [6–8]. Evidence that value learning is implemented by the brain emerged from early foundational work on primates indicating that brainstem dopaminergic neurons instantiate prediction errors—differences between actual and expected reward—that are well suited for algorithmic implementations of value learning [9]. Since then, many human brain imaging studies have shown that activation in the orbitofrontal cortex (OFC) and other regions is correlated with expected value during reward learning and other types of economic decision-making tasks [5,10,11].

New evidence, however, has cast into question whether humans indeed learn absolute expected values or may be instead learning relative preferences among limited sets of options. Two recent studies showed that people's choices reflect relative preferences because when they are rewarded for choosing one out of two options, they do not only form a preference in favor of the option they chose, but also a preference against the option they did not choose [12,13]. Neural data reveal a similar picture. Neural firings in areas considered to encode value such as the OFC and the striatum have been found to encode normalized values that, in fact, have no absolute meaning and can only be interpreted as relative preferences compared to other options sampled in the same context [14,15]. Such relative preference encoding is evident even when each individual option is encountered separately [16,17]. These studies among others [18–25], have led researchers to propose alternative models of learning, according to which humans learn preferences between options without encoding the absolute value of each option [7,8,26–29].

Here, we test a novel hypothesis that humans flexibly adapt the degree to which they form absolute expected values and relative preferences based on the opportunities and incentives afforded by the environment. It is well established, across a wide range of machine learning applications, that learning environments that provide a more diverse set of learning exemplars aid generalization of learned information to new input patterns and unfamiliar contexts [30,31]. In the case of learning to maximize reward, the set of exemplars corresponds with the set of possible options, and learning about a broader set of options could make it more clearly evident that the value of an option does not depend on the other options it is pitted against—that is, that each option has an absolute value. Additionally, the broader the set of options, the greater is the space of possible choice sets (i.e., a choice set is a set of simultaneously available options from which one chooses), some of which have yet to be encountered. The prospect of having to choose among novel sets of previously encountered options makes it worthwhile to

form preferences that can be used to choose among such sets, which is precisely what absolute values are best suited for. By contrast, relative preferences produce suboptimal choices among unfamiliar choice sets, since they only encode how valuable options were relative to the other options they were previously pitted against [12,27].

These considerations suggest at least two types of training diversity may support and incentivize value learning. The first type of diversity relates to how many options a person learns about concurrently within a given learning session (henceforth, *concurrent diversity*), whereas the second type of diversity is the number of alternative options a given option is cumulatively pitted against, across all learning sessions (henceforth, *cumulative diversity*). These two types of diversity are dissociable since an option can be learned concurrently with fewer other options, yet across multiple learning sessions it may be cumulatively pitted against more other options.

To test the impact of concurrent and cumulative diversity on the formation of absolute values, we designed a novel multi-day reward learning experiment comprising twenty learning sessions. In each session, subjects' goal was to maximize their reward by choosing among pairs of images, each of which was associated with a fixed probability of reward. The probabilities were not told to subjects and could only be learned through trial and error, by observing the reward outcomes of chosen images. To manipulate concurrent diversity, we varied how many images subjects learned about concurrently in each learning session. To manipulate cumulative diversity, we varied the total number of other images each image was pitted against over two separate learning sessions. Critically, the multi-session design allowed us to assess the formation of absolute values by asking subjects to choose between images that were never directly paired together during learning. To enhance the distinction between absolute values and relative preferences, we had images with the same reward probability learned against other images with either mostly lower or mostly higher reward probabilities. An absolute value learner would have no preference among these images, whereas a relative preference learner would prefer the option that ranked higher in its original learning context.

## Results

27 subjects (ages 20 to 30; Mean = 24 ±.5) completed two learning sessions a day of a reward learning task over a period of ten days (Fig 1). In low-concurrent-diversity sessions, subjects learned about three images at a time, whereas in high-concurrent-diversity sessions, subjects learned about six images at a time. Every image appeared in two learning sessions, but low-cumulative-diversity images were pitted against the same images in both sessions whereas high-cumulative-diversity images were pitted against different images.

### Subjects formed preferences in favor of more rewarding images

To validate our task, we first determined whether subjects successfully learned to choose images associated with higher reward probabilities. Subjects indeed tended to choose the more rewarding image out of each pair, doing so with 86% (SEM ±1%) accuracy during learning (Fig 2A; chance performance = 50%). As can be expected, subjects' performance was lower in the conditions that required learning about more images (i.e., high concurrent diversity; Fig 2B, left panel) or about more pairs of images (i.e., high cumulative diversity; Fig 2B, right panel). However, subjects performed considerably above chance in all conditions, and showed gradual improvement with each new image as they tried out choosing it and observed its outcomes (Fig 2C). These results confirm that the task was effective in getting subjects to form preferences among images based on how often each image was rewarded.

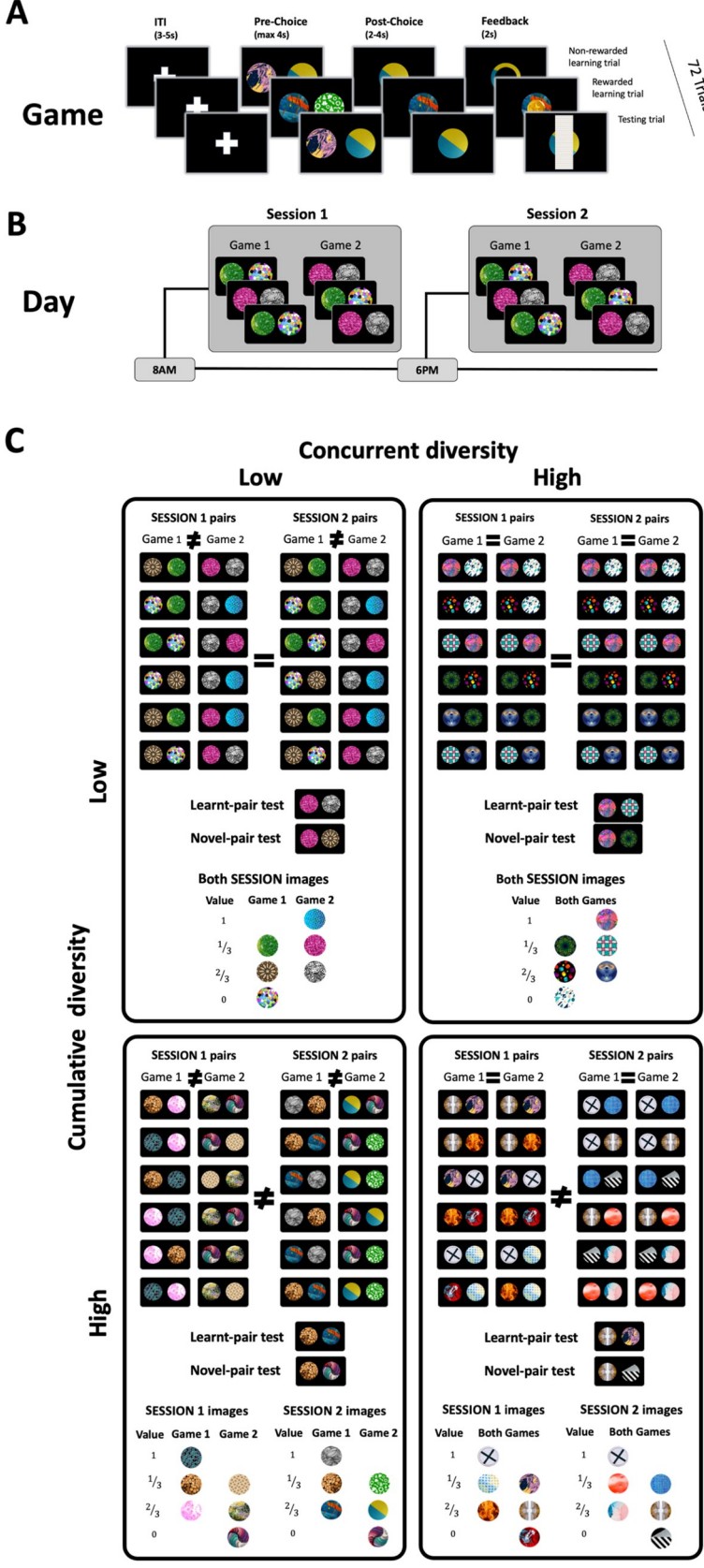

**Fig 1. Experimental design. (A) Reward learning game.** On each trial, subjects were asked to choose one of two circular images. Following their choice, subjects either received or did not receive a reward of 1 coin based on a fixed reward probability associated with the chosen image (0, 1/3, 2/3, or 1). Each game consisted of 48 such 'learning' trials, interleaved with 24 'testing' trials wherein subjects chose between images about which they learned in prior sessions. Outcomes were not revealed in testing trials to prevent further learning. Every image first appeared in 64 learning trials over two sessions before subjects were tested on it. ITI: inter-trial-interval. **(B) Daily schedule.** Each day, subjects performed two experimental sessions on a specially designed mobile phone app [32] one in the morning (on average, at 8:56 am, and no earlier than 6:00 am) and one in the evening (on average, at 6:12 pm, and no earlier than 4:00 pm). In each session, subjects played two games in which they learned about a total of six images. **(C) Experimental conditions.** Four experimental conditions were implemented along 10 days of learning, each lasting two to three days. Each condition is illustrated via a representative selection of six pairs of images subjects chose between in two different sessions. Within the low concurrent diversity condition (left two columns), three images were learned over the span of each game, whereas in the high concurrent diversity (right two columns) six images were learned over the span of two games. Thus, in both conditions, each image appeared in 32 learning trials per session. In the low-cumulative-diversity conditions, each image was pitted against the same two images in two consecutive learning sessions, whereas in the high-cumulative-diversity conditions, images were pitted against different images (sometimes in two nonconsecutive sessions; see Methods for details). Three days of training, as opposed to two, were required for high-cumulative-diversity conditions so that each image could be pitted against different images in its two learning sessions. Conditions were randomly ordered, and equal in terms of average reward probability and number of images learned per session. Testing trials involved choosing between two images the subject already chose between during learning (Learned-pair trials, 25% of testing trials) or novel pairings of images learned separately (Novel-pair trials, 75% of testing trials). On average, pairs were tested 27 ±.8 times.

## Concurrent diversity increased generalization

A hallmark of value learning is the ability to generalize learned preferences to novel settings. To test generalization, we had subjects choose between images that had not been previously pitted against each other ('novel pair' testing trials). We compared subjects' accuracy on these trials to accuracy in choosing between images that subjects had encountered during learning ('learned pair' testing trials). Novel and learned pairs involved the same images and presented subjects with similarly difficult choices, in the sense that the time elapsed since learning was roughly the same (novel pairs = 1.88 days ±.08, learned pair = 1.75 ±.08 following learning), as was the average difference in reward rate between the two images that made up a pair (reward rate difference: $\Delta_{novel-pair}$ = 48.4% ±2%, $\Delta_{learned-pair}$ = 49.1%±3%). However, only novel pairs presented subjects with choices between images learned in different games with different sets of other images. This presents no challenge to an absolute value learner, but a relative preference learner might end up choosing an image with a lower expected value simply because it was learned against worse images (and thus acquired a higher relative value). For this reason, a pure absolute value learner can be expected to perform equally well in choosing between novel and learned pairs, whereas a relative preference learner should perform worse in choosing between novel pairs.

We found that subjects successfully chose the image with the higher reward probability in 83% of novel-pair trials (SEM ±2%; chance performance = 50%). This level of accuracy, however, was significantly lower than the accuracy subjects demonstrated on learned-pair trials (Mean = 87% SEM ±3%; bootstrap p = .03). Subjects thus generalized their preferences well, but did not do so perfectly.

We therefore asked whether success in generalization was affected by the diversity of learning experiences. To quantify generalization, we computed the drop-off in accuracy from learned-pair to novel-pair testing trials. Since we found no interaction between the effects of concurrent and cumulative diversity (p = .86 bootstrap test), we separately examined each while marginalizing over the other. Strikingly, we found that accuracy did not significantly differ between novel-pair and learned-pair trials for images learned in conditions of high concurrent diversity (Fig 3; Mean = -1% ±2%; p = .68, bootstrap test). Conversely, for low-concurrent-diversity images, subjects performed substantially worse in choosing between

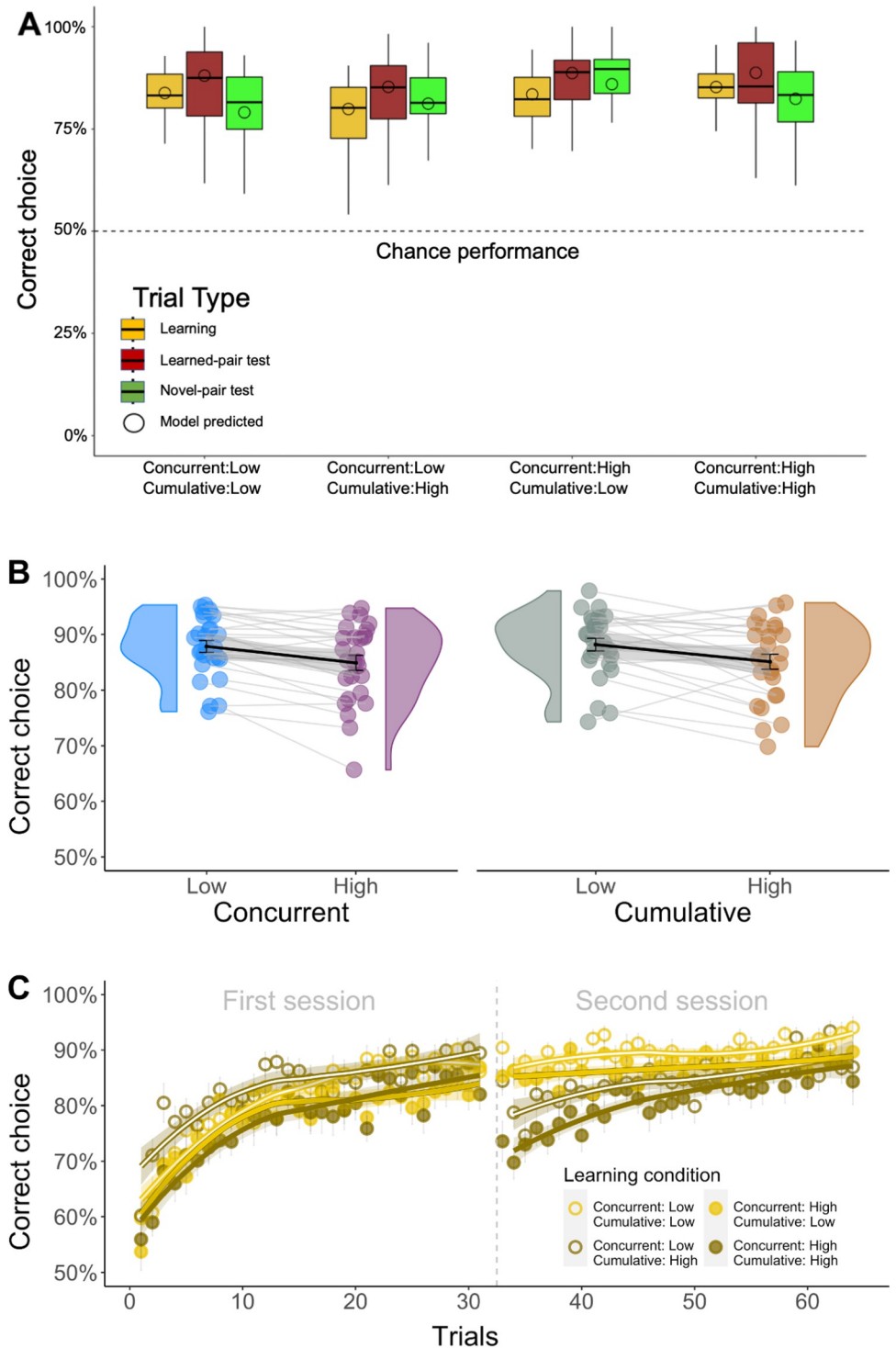

**Fig 2. Overall performance.** *n* = 27 subjects. **A) Choice accuracy as a function of trial type and condition.** A choice was considered accurate if the subject selected the image with the higher reward probability. Subjects performed significantly above chance (50%) in all trial types and conditions (CI = [.834,.879]). The plot shows total (vertical lines) and interquartile (boxes) ranges and medians (horizontal lines). Also shown are mean accuracies predicted by a computational model that was fitted to subjects' choices (circles; see details under *Computational Formalization* below). **B) Effect of training diversity on accuracy during learning.** Accuracy was higher in sessions with low (91% SEM ±1%) compared to high (87% SEM ±1%) concurrent diversity (p$_{corrected}$ = .048, bootstrap test), and trended higher in sessions with low (90% SEM ±1%) compared to high (88% SEM ±1%) cumulative diversity (p$_{corrected}$ = .078,

bootstrap test). The plot shows individual subject accuracy (circles), group distributions of accuracy levels (violin), group means (thick lines) and standard errors (gray shading). **C) Learning curves for each experimental condition.** Accuracy in trials involving a given image as a function of how many trials the image previously appeared in. A drop-off in accuracy can be observed for high-cumulative-diversity images (dark) at the beginning of the second session, because these images were then pitted against new images. The plot shows group means (circles), standard errors (vertical lines), local polynomial regression lines ([33]; curves) and confidence intervals (shading). Given that performance was always above chance (50%), y-axes in panels B and C focus on this range.

novel pairs (Mean = -7% ±2%; $p_{corrected}$ = .004 bootstrap test). This difference between low and high concurrent diversity was neither due to a difference in learned-pair trials nor in novel-pair trials (S1 Table), but specifically reflected the drop-off in accuracy between them ($p_{corrected}$ = .015 bootstrap test).

In contrast, cumulative diversity did not impact the performance drop-off from learned-pair to novel-pair trials (-3% ±2% vs -4.8% ±2% $p_{corrected}$ = .29 bootstrap test). This was despite the fact that pairs of images were encountered during learning half as many times in

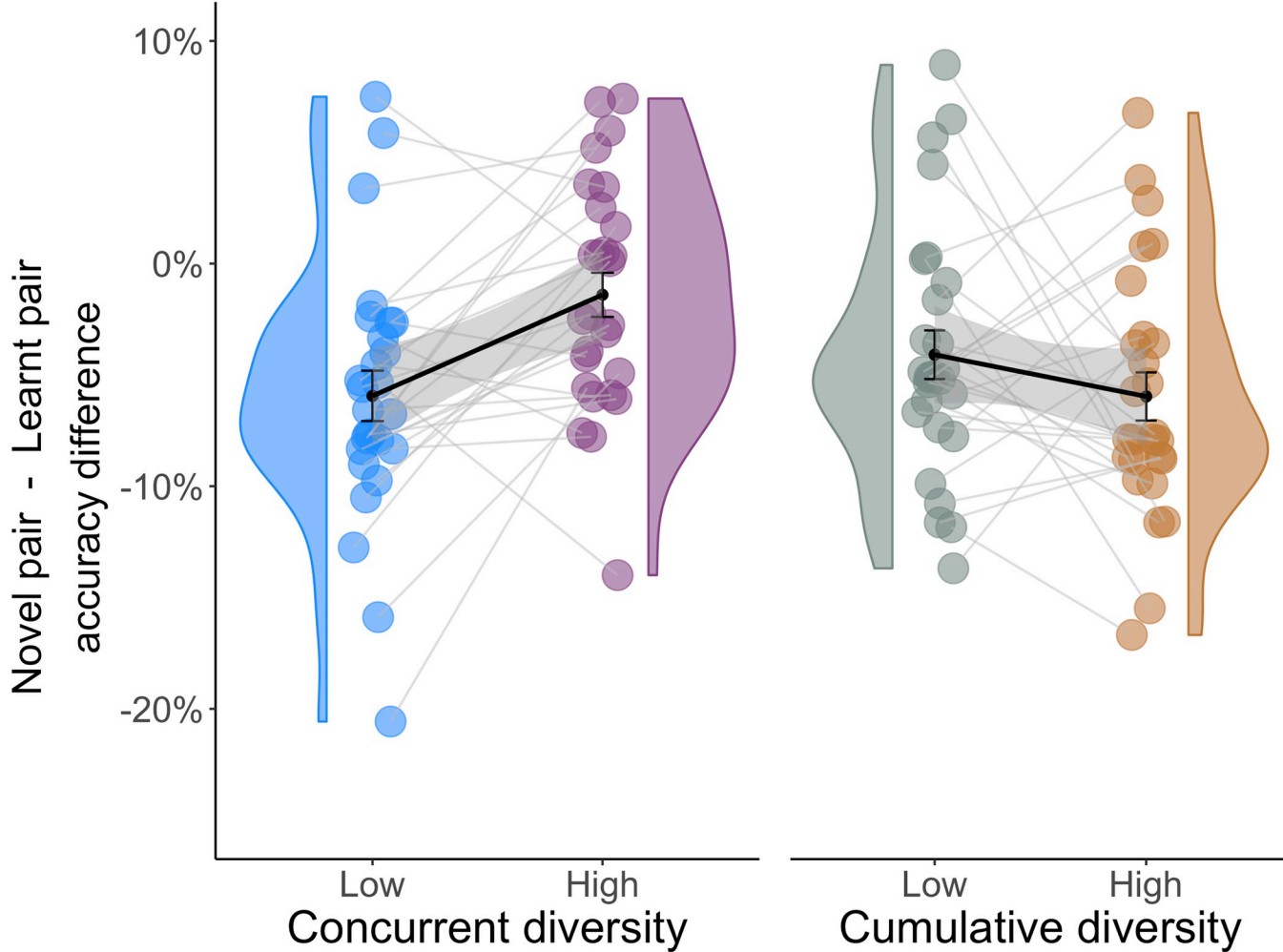

**Fig 3. Generalization performance.** *n* = 27 subjects. Drop-off in accuracy in novel-pair compared to learned-pair testing trials, as a function of training diversity. Concurrent diversity had a significant effect on this measure of generalization ($p_{corrected}$ = .004, bootstrap test) whereas cumulative diversity did not significantly affect it ($p_{corrected}$ = .29, bootstrap test). The plot shows individual subject accuracy (circles), group distributions of accuracy levels (violin), group means (thick lines) and standard errors (gray shading).

conditions of high cumulative diversity. For this reason, we expected that learned-pair performance would be compromised by cumulative diversity, and this was indeed the case (as evident by comparing accuracy on learned-pair trials to the level of accuracy achieved at the last 5 trials of learning; $Mean_{low}$ = -2.3% SEM ±1%, $Mean_{high}$ = -4.2% ±1%, p = .03 bootstrap test). However, accuracy on novel-pair trials was similarly compromised by cumulative diversity ($Mean_{low}$ = -5.6% ±2%, $Mean_{high}$ = -7.9% ±2%, p = .026 bootstrap test), and thus no benefit to generalization was observed.

These results show that increasing the number of options about which a person *concurrently* learns improves their ability to generalize their learned preferences to novel choice sets.

## Concurrent diversity reduces influence of other options' outcomes

The observed improved generalization suggested that concurrent diversity enhanced absolute value learning. To further investigate this possibility, we examined another key consequence of absolute value learning, namely, that the preferences it forms depend only on the available images' prior outcomes. By contrast, relative preferences also account for the outcomes of other images against which the presently available images were pitted during learning. These latter outcomes determined how each of the available images ranked compared to other images during learning. Due to accounting for these outcomes, a relative preferences learner should show no preference between images with similar rankings during learning even if their absolute values differ, but favor similarly rewarded images that ranked higher in their original learning context (i.e. a rank-bias).

In examining choices between similarly ranked images with different absolute values, we found that concurrent diversity improved subjects' accuracy (S1 Fig; Mean_high = .83±.02; Mean_low = .78±.02; $p_{corrected}$ = .035 bootstrap test), as consistent with a shift towards value learning. By contrast, cumulative diversity impaired performance on such trials (Mean_low = .82±.02 Mean_high = .77±.02 $p_{corrected}$ = .02 bootstrap test), as consistent with its general detrimental effect on overall performance.

Visualizing choices between similarly rewarded images (i.e. less than 10% difference in reward rate) showed that subjects preferred images that ranked higher during learning (i.e., that were pitted against images with lower reward probabilities) under low concurrent diversity ($Mean_{low}$ = 63% SEM ±6%, with 50% representing no preference between images) but not under high concurrent diversity ($Mean_{high}$ = 49% SEM ±5%; Fig 4A). This difference between conditions trended towards significance (p = .06 permutation test) and was not evident as a function of cumulative diversity (Mean low = 57% ±7%, Mean high = 48% ±8%).

The above measure of rank bias, however, is limited in both sensitivity and validity. First, it ignores the random differences that inevitably exist in the actual outcomes of similarly ranked images. Second, it does not utilize the information that exists in subjects' choices between differently ranked images. Third, it is confounded by the fact that higher ranking images were chosen more during learning, since they were pitted against less rewarding images. This latter confound is important because it means that more outcomes were observed for higher-ranking images, which may have allowed subjects to develop greater confidence regarding their value.

To address these challenges, we used a Bayesian logistic mixed model that predicted subjects' choices based on differences between the currently available images' own reward history ($\beta_{Own}$; i.e., proportion rewarded), the number of times each image was chosen ($\beta_{Times\ Chosen}$ i.e., sampling bias), and the reward history from trials in which the currently available images were rejected in favor of other images (Fig 4B). The latter was separated into two separate regressors, for prior outcomes of rejecting a presently available image in favor of the other presently available image ($\beta_{Current\ alternative}$) and of rejecting it in favor of any other image

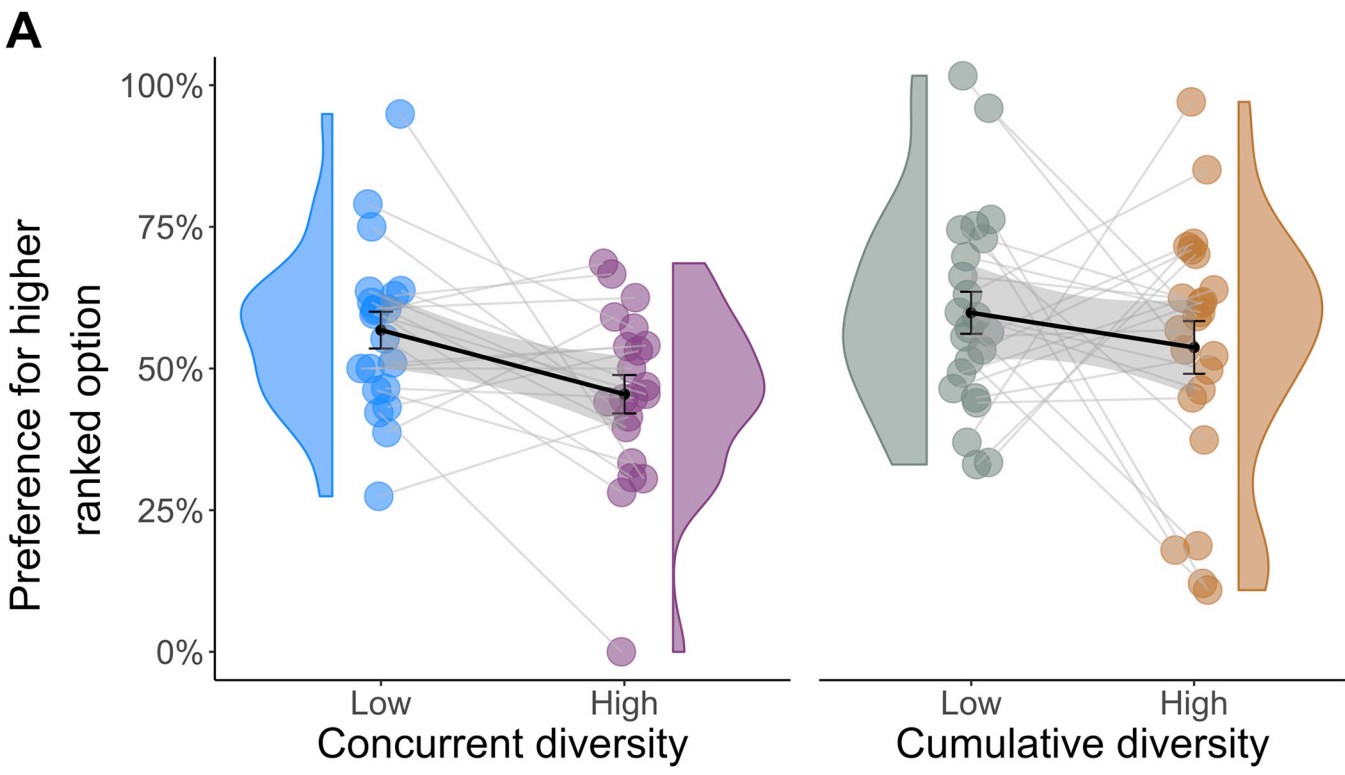

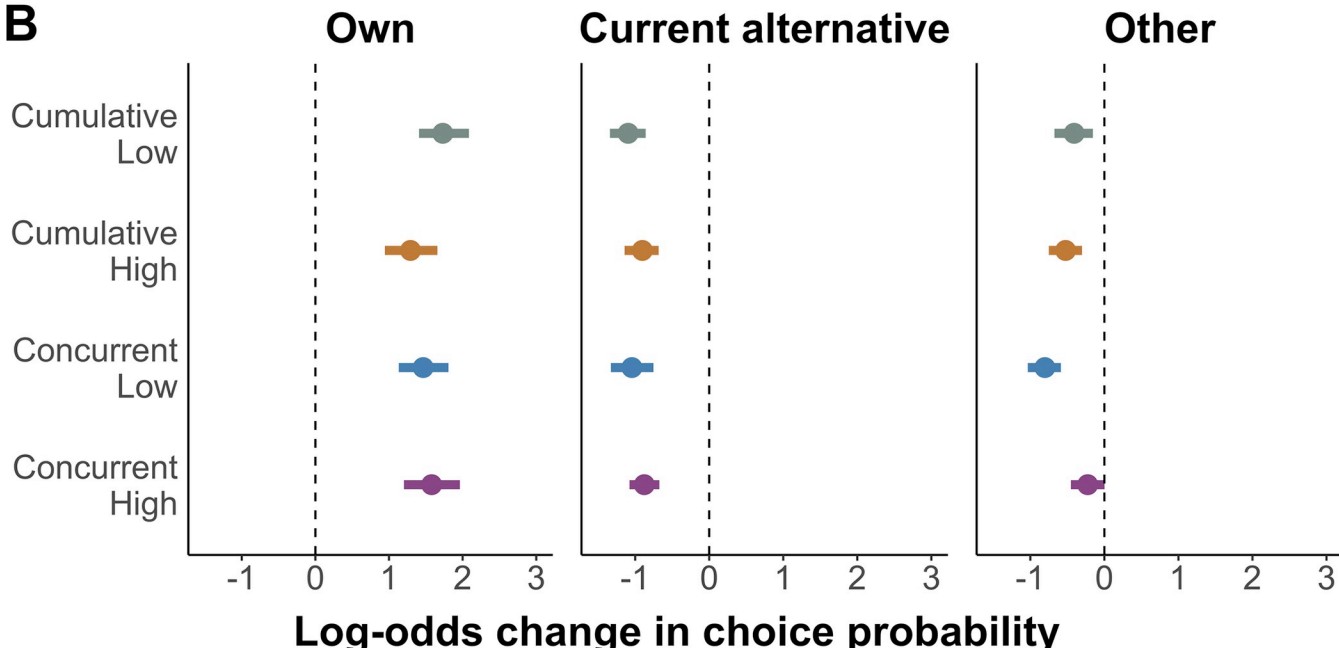

**Fig 4. Effect of other options' outcomes. A) Rank bias as a function of training diversity.** Y-axis shows the percent of trials involving similarly rewarded images in which subjects chose the image that ranked higher during learning. A subject's image rankings were based on how many times the subject chose each image over the other images it was pitted against (best, second-best, or worst, see Methods for further details). The plot depicts individual subjects' choices (circles), group distributions (violin), group means (thick lines) and standard errors (gray shading). **B) Effect of reward histories on choice.** The plot shows the log odds effect on choice of three types of reward. Own: differences in reward history of currently available images. Current alternative: differences in reward history when rejecting one available image in favor of the other available image. Other: differences in reward history when rejecting one of the available images in favor of any other image.

($\beta_{\text{Other}}$). Importantly, only the latter regressor unequivocally captures the effect of other images' outcomes that are irrelevant for maximizing absolute expected value. To determine whether the effects of different types of outcomes were modulated by training diversity, we included regressors for concurrent and cumulative diversity and interactions between both types of diversity and each type of reward history.

The results confirmed that in addition to the strong impact of an image's own reward history ($\beta_{\text{Own}} = 1.97$ [1.63, 2.31]), preference for an image was inversely influenced by the outcomes of both the current alternative ($\beta_{\text{Current alternative}} = -.43$ [-.64, -.21]) and of the other images it had previously been pitted against ($\beta_{\text{Other}} = -.53$ [-.75, -.32]). Thus, the more subjects were rewarded when not choosing an image, the less likely they were to prefer it on subsequent trials. Most importantly, the influence of other images' reward history was reduced by high concurrent diversity ($\beta_{\text{Other×Concurrent}} = .39$ CI = [.30, .49]). No additional interactions were found (S2 Table), except for an interaction of cumulative diversity with the impact of an image's own outcomes, as consistent with cumulative diversity's general detrimental effect on performance. Thus, concurrent, but not cumulative, diversity reduced the influence of other options' rewards that are irrelevant for inferring absolute value.

Finally, to inquire whether the effect of cumulative diversity was indeed specific to trials that exclusively probed absolute value, we examined subjects' accuracy in novel-pair testing trials wherein one of the images had both higher absolute and higher relative value than the other image. Choosing correctly on such trials does not require absolute values. As expected, we found no significant effect of concurrent diversity in these trials ($p_{\text{corrected}} = .22$). Here, too, we found a trend for subjects to perform worse on images learned in high, as compared to low, cumulative diversity ($p_{\text{corrected}} = .052$ bootstrap test). Taken together, the results indicate that concurrent diversity specifically improved accuracy on trials that required absolute values to choose correctly.

## Computational formalization of value and preference learning

Our results evidenced signs of both absolute value and relative preference learning. On one hand, subjects successfully learned reward maximizing choices and generalized well to novel choice sets, as consistent with absolute value learning. On the other hand, subjects performed still better at choosing among familiar choice sets, and they preferred images that were relatively more valuable in their original learning context, as consistent with relative preference learning. Critically, learning about more images concurrently diminished or even eliminated the signs of relative preference learning.

We next tested whether this set of results can be coherently explained as reflecting the operation of two learning processes—absolute value and relative preference learning—the balance between which changes as a function of concurrent diversity. To do this, we fitted subjects' choices during both learning and testing with a computational model that combines value and preference learning (as proposed by [26]). We then examined the best-fitting values of the model's parameters to determine the degree to which value and preference learning were each employed in each experimental condition.

To formalize absolute value learning, the model represents subject beliefs about the absolute values of images as beta distributions, defined by two parameters $a_i$ and $b_i$. This beta distribution represents the reward probability that is believed to be associated with each image given the outcomes obtained for choosing it. Thus, $a_i$ and $b_i$ accumulate the number of times that choice of image $i$ was rewarded and not rewarded:

$$a_i \leftarrow \gamma a_i + O \tag{1}$$

$$b_i \leftarrow \gamma b_i + (1 - O) \tag{2}$$

where $O = 1$ if the choice was rewarded and $O = 0$ if it was not. Here, $\gamma$ serves as a leak parameter allowing for the possibility that more recent outcomes have a greater impact on subjects' beliefs ($\gamma = 1$ entails that outcomes are equally integrated, whereas $\gamma < 1$ entails overweighting of recent outcomes). The decision variable provided by this form of learning is the absolute value ($V$) of image $i$, which is estimated as the expected value of the image's beta distribution:

$$V(i) = \frac{a_i}{a_i + b_i} \tag{3}$$

To isolate the key computation distinguishing relative preference learning from value learning, we use the same learning rules defined above to generate a relative preference $W(i,j)$, for image $i$ over image $i$, that accounts for all outcomes observed for choosing between the two images. When applying Eqs 1 and 2 for preference learning, $O = 1$ if image $i$ was chosen and rewarded or if image $j$ was chosen and not rewarded, and $O = 0$ if image $j$ was chosen and rewarded or if image $i$ was chosen and not rewarded. To enable preference learning to exhibit preferences among previously unencountered pairs of images, a general relative preference $W(i)$ was computed for each image $i$ by accumulating in the same fashion the outcomes of choosing image $i$ or the other image, across all learning trials involving image $i$. Accounting for the outcomes of the other images distinguishes the relative preference $W(i)$ from the absolute value $V(i)$.

When facing a choice between image $i$ and image $j$, the probability that the model will choose either image is computed based on a weighted sum of the images' absolute value and relative preference:

$$P(\text{choice} = i) = \frac{e^{\beta_{\text{value}} V(i) + \beta_{\text{preference}} W'(i)}}{e^{\beta_{\text{value}} V(i) + \beta_{\text{preference}} W'(i)} + e^{\beta_{\text{value}} V(j) + \beta_{\text{preference}} W'(j)}} \tag{4}$$

where $W'(i)$ is itself a weighted sum of $W(i)$ and $W(i,j)$, with a free parameter controlling their relative weights. Importantly, $\beta_{\text{value}}$ and $\beta_{\text{preference}}$ are distinct inverse temperature parameters, the respective magnitudes of which determine the degree to which absolute values and relative preferences influence choice. Thus, the values of these parameters that best fit subjects' choices can be used to quantify the degree to which value and preference learning manifested in each experimental condition [34,35]. To this end, we allowed the two inverse temperature parameters to vary as a function of concurrent and cumulative diversity, for either learning or testing trials.

## Subjects combine value and preference learning

To determine whether a combination of value and preference learning was needed to explain subjects' choices, we compared the full model to two sub-models, one that only learns absolute values ($\beta_{\text{preference}} = 0$) and one that only learns relative preferences ($\beta_{\text{value}} = 0$), as well as to a number of additional alternative learning models (see Methods). We found that the full model accounted for subjects' choices across both learning and testing trials significantly better than the alternative models (Fig 5 and S5 Table). Moreover, only the full model was able to recreate in simulation all the behavioral findings, including generalization performance, effect on choice of outcomes for other images, and rank bias (see S2 Fig and Falsification of alternative models in Methods).

## Concurrent diversity enhances value learning

Examining the values of the parameters that best-fitted subjects' choices across all trials showed that value learning generally predominated over relative preference learning ($\beta_{\text{value}} = 4$

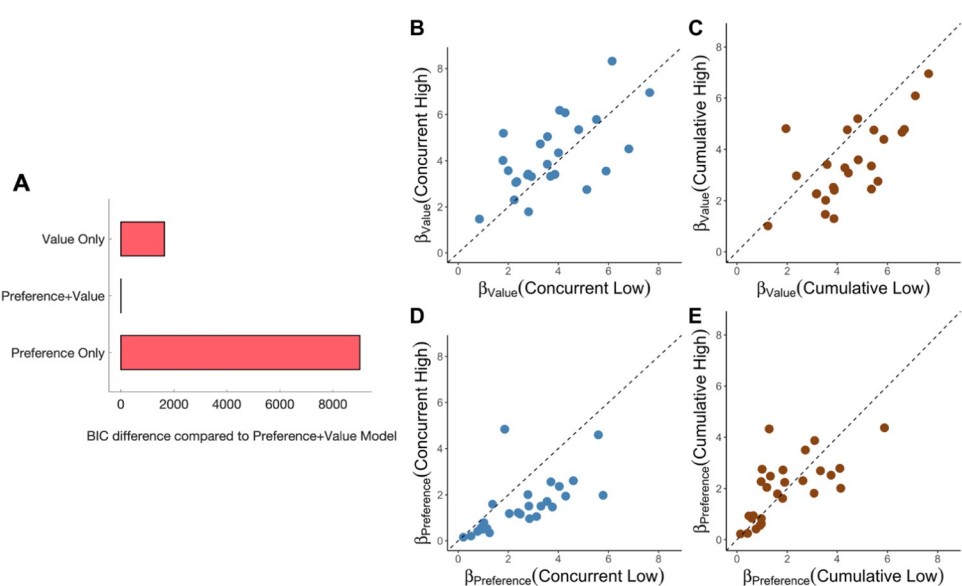

**Fig 5.** *n* = **27 subjects. A) Model comparison.** Comparison of the combined Preference+Value model to models that learn only absolute values or relative preferences. The models are compared by means of the Bayesian Information Criterion (BIC; [36]). Lower BIC values indicate a more parsimonious model fit. **B) Modeled Utilization of Value and Preference Learning.** Individual subject model parameter fits showing the effects of concurrent and cumulative diversity on the degree to which preference and value learning manifested in subjects' choices. Each dot represents a subject. Dashed lines mark where utilization of the form of learning is equal for low and high diversity.

±.28 vs $\beta_{preference}$ = 1.88 ±.22), as befitting a task that involves frequent choices between options from different learning contexts. Importantly, however, preference learning manifested to a greater extent in conditions of low concurrent diversity (low concurrent: $\beta_{preference}$ = 2.6 ±.29; high concurrent: $\beta_{preference}$ = 1.44 ±.21; *p*<.001 permutation test), whereas value learning manifested to a greater extent in conditions of high concurrent diversity (low concurrent: $\beta_{value}$ = 3.84±.31; high concurrent: $\beta_{value}$ = 4.33±.2 *p*<.001 permutation test).

By contrast to concurrent diversity, cumulative diversity inhibited value learning (low cumulative: = 4.7 ±.5; high cumulative: $\beta_{value}$ = 3.5 ±.4; p < .001 permutation test) and had no significant impact on preference learning (low cumulative: $\beta_{preference}$ = 1.89 ±.13; high cumulative: $\beta_{preference}$ = 1.92 ±.1; p = .32 permutation test). These results indicate that concurrently learning about a broader set of options enhances the use of absolute values for making choices.

## Testing alternative interpretations

A necessary consequence of varying concurrent diversity is an extension of the duration of learning, since twice as many trials are required to learn about twice as many images. This raises the possibility that it was simply the duration of learning, and not the diversity of learning exemplars, that shifted learning from relative preferences to absolute values. To test this interpretation, we implemented a variant of our model where a shift towards absolute-value learning progresses gradually during learning irrespective of concurrent diversity. We compared this model to a matched implementation of a concurrent diversity effect, that is, where a gradually developing shift progresses towards value learning under high concurrent diversity but towards preference learning under low concurrent diversity. Model comparison favored the latter model over all other models (ΔBIC +741). This result indicates that shifts between preference and value learning indeed developed gradually. Most importantly, though, this result confirms again that the direction of the shift depended on concurrent diversity.

A second necessary consequence of concurrently learning about more images is that consecutive presentations of an image will be separated by more intervening trials. Larger separation might itself affect the predominant form of learning, either because values and preferences decay during the intervening trials at different rates, or because a larger separation between outcomes affects the degree to which subjects overweight recent outcomes in forming values and preferences. To test the first possibility, we modified our model so as to allow values and preferences to decay during the intervening trials (via the leak parameters $\gamma_{value}$, $\gamma_{preference}$). This model fitted the data worse (ΔBIC +220), ruling out decay during intervening trials. To test the second possibility, we modified our model so that the overweighting of recent outcomes (also controlled by the leak parameters) could vary as a function of diversity conditions. This model too fitted the data substantially worse (ΔBIC + 350).

Finally, we examined an alternative hypothesis that participants make use of the transitivity of relative preference, thereby inferring a global rank of items without learning the expected value of each item. However, even among pairs of images with no transitive relation between them, subjects were significantly above chance in selecting the higher value image (*Mean* = .82±.1). Moreover, the effects of concurrent diversity on generalization were significant within this subset of trials as well ($p_{corrected}$ = .003 bootstrap test).

Thus, neither the duration of learning, nor the presumed effects of interleaving trials, nor transitive inferences offer a successful alternative explanation for the enhancement of absolute value learning by high concurrent diversity of learning exemplars.

## Discussion

We found that increasing the number of options a person concurrently learns about shapes reward learning in several ways. It first reduces performance during learning, but then leads to more successful generalization, removes a bias in favor of options that ranked higher during learning, and generally decreases the degree to which preference for an option is influenced by presently irrelevant options' outcomes. Computational modeling shows that all of these effects are coherently explained by a shift away from relative preference and towards absolute value learning. These findings offer a meaningful extension of previous demonstrations of absolute value [1,2,5] and relative preference [12,13,27] learning in humans, by identifying key conditions under which the former is diminished in favor of the latter, namely, conditions of high concurrent training diversity.

The enhancement of absolute values and inhibition of relative preferences that we observed can best be understood in light of past suggestions that encoding context-specific information aids performance as long as the agent remains within the learning context, but is ill suited for generalizing policies to other learning contexts [37,38]. Relative preference learning is inherently specific to the learning context and impairs generalization to novel choice sets. Our findings show that such context-specific learning is promoted by a learning experience that limits the possibility of encountering novel choice sets, specifically, by reducing the number of options. In this sense, the shift between preference and value learning in our experiment can be thought of as a rational adaptation. This perspective is supported by a recent finding that value learning is enhanced by expectations of having to choose between options from different learning contexts [39]. Here, though, we demonstrate that absolute value learning can be enhanced even absent a direct manipulation of the need to choose between options from different learning contexts. Increasing the number of options is sufficient for this purpose. Conversely, with a low number of options, relative-preference learning remains clearly evident despite subjects being aware of the need to choose across contexts.

Our findings agree with prior work showing that emphasizing comparisons between a limited number of specific images, for instance by repeatedly presenting subjects with a choice

between the same two options and providing reward information about the foregone option, promotes learning of relative values [27]. However, the process by which the formation of relative values in the latter experiments has so far been explained–namely, normalization to the range of outcomes experienced during learning–cannot explain relative preference learning in our experiment. This is because the range of outcomes in our experiments was the same in all learning sessions. By contrast, the model we proposed here for relative preference learning may coherently account for both our findings and the findings that had previously been attributed to normalization.

Though both concurrent and cumulative diversity increased task difficulty, as evident by poorer performance during learning, cumulative diversity did not have the effect of improving generalization. This result has two key implications. First, it contradicts previous suggestions that it is task difficulty per-se that promotes absolute value learning [27]. Second, it suggests that the formation of absolute values is not promoted by the global diversity of learning exemplars encountered during the entire course of learning, but rather, by the local diversity that characterizes the immediate learning context.

We successfully ruled out several alternative explanations for the finding that concurrent diversity promotes absolute value learning, including some possible effects of increased duration of learning and greater separation between consecutive choices of an image, both of which are direct consequences of concurrently learning about more images. Another important consequence of such learning, which we have not addressed here, is increased working memory load [40]. Future experiments could disentangle the effects of number of images and working memory load by introducing unrelated tasks during learning, so as to increase working memory load without changing the number of images about which subjects concurrently learn.

Another open question remains as to a full functional description of the relationship between concurrent diversity and absolute value learning. Our model, which was tested on three (low diversity) or six (high diversity) concurrently learned images does not allow us to extrapolate to learning with other set sizes. Clarifying the full functional relationship between diversity and value learning can be aided by extending the current experimental approach to testing additional levels of diversity, as well as by further developing a mechanistic understanding of how diversity promotes value learning.

Several studies have investigated the neural basis of value [5,9,10,11] and preference [12,14,15] learning in isolation, and the potential instantiation of relative preferences via sampling from memory during choice [41–45]. However, it is yet unknown how the brain arbitrates between preference and value learning. One relevant line of work comprises studies on how concurrent diversity influences the brain regions recruited for learning [40]. Though this work has not examined absolute values and relative preferences, it has shown that increasing the number of items people concurrently learn about strengthens activation in a striatal-frontoparietal network implicated in value learning. Future studies could investigate the involvement of this network and other regions in arbitrating between value and preference learning as environmental conditions change.

## Conclusion

Our findings contribute to the ongoing debate concerning the extent to which people learn absolute values versus relative preferences. We show that absolute-value learning depends on a characteristic of the immediate learning context, namely, the diversity of learning experiences it offers. We find that increased diversity, despite impairing performance in the short term, has the effect of enhancing learning of absolute values which generalize well to novel contexts.

Such generalization is essential for making decisions in real life where our experiences are inevitably fragmented across many different contexts.

## Contact for resource sharing

Further information and requests for resources or raw data should be directed to and will be fulfilled by the Lead Contact, Levi Solomyak (levi.solomyak@mail.huji.ac.il).

## Methods

### Ethics statement

The experimental protocol was approved by the Hebrew University local research ethics committee, and written informed consent was obtained from all subjects.

### Subjects

27 human subjects (14 male,13 female), aged 20 to 30 (Mean = 24 SEM ±.5), completed the experiment which consisted of 3556 trials [46]. Given the size of the dataset obtained for each subject (an order of magnitude greater than in typical learning experiments) and the effect size found in similar prior literature [27], we expected that a meaningful finding would manifest as at least a large effect (Cohen's D = .8; [47]). We thus selected a sample size that would provide at least 80% power of detecting such an effect (i.e., n> = 26). The experiment was discontinued midway for 3 additional subjects due to failure to complete learning sessions or evidence of random choosing. Subjects were recruited from a subject pool at Hebrew University of Jerusalem as well as from the Jerusalem area. Before being accepted to the study, each subject was queried regarding each of the study's inclusion or exclusion criteria. Inclusion criteria included fluent Hebrew or English and possession of an Android smartphone that could connect to wearable sensors via Bluetooth Low Energy. Exclusion criteria included age (younger than 18 or older than 40), impaired color discrimination, use of psychoactive substances (e.g., psychiatric medications), and current neurological or psychiatric illness. Subjects were paid 40 Israeli Shekels (ILS) per day for participation and 0.25 ILS for each coin they collected in the experimental task, which together added up to an average sum of 964 ±42 ILS over the entire duration of the study.

Subjects who missed two sessions of the experiment or who displayed patterns of making random choices were automatically excluded from the study. Random choosing was indicated by chance-level performance or reaction times below 1000 ms, which our previous experience [32] suggested is consistent with inattentive performance.

### Experimental design

To test for value and preference learning, we had subjects perform a trial-and-error learning task over a period of 10 days. On each trial, subjects chose from one of two available images, and then collected a coin reward with a probability associated with the chosen image. Each game included 48 such learning trials involving a set of 3 images with reward probabilities of either {0, .33,.66} or {.33, .66 and 1}. These probabilities were never revealed to the subjects. Subjects were only instructed that each image was associated with a fixed probability of reward. Subjects played four games a day, two in a morning session and another two in an evening session. Over a total of 20 sessions, subjects learned about 60 unique images, each appearing in 64 learning trials over two sessions.

To assess whether concurrent or cumulative diversity promotes absolute value learning, we tested subjects on four experimental conditions involving either low or high levels of each type of diversity. Task conditions were randomly ordered across days in order to avoid confounds

related to fatigue or gradual improvement in learning strategy. Images learned in low cumulative diversity conditions were learned over the span of two consecutive days. To satisfy the constraints of high cumulative diversity concerning which images are pitted against which, high cumulative diversity conditions spanned three days (see below). All four conditions yielded the same expected payout, since the average reward probability associated with images within each condition was .5.

To enhance the distinction between absolute values and relative preferences, we had images with the same absolute value (i.e., equal reward probability) learned against other images with mostly lower reward probabilities (i.e., in games where the probabilities were {0, .33,.66}; low reward context) or mostly higher reward probabilities (i.e., in games where the probabilities were {.33, .66 and 1}; high reward context). Within each day, an equal number of images were learned in the high and low reward contexts.

## Concurrent diversity

**Low**: In these conditions, the two games within each learning session were independent of one another, each involving a distinct set of three images, with each trial randomly pairing two of the three images. Thus, the number of images subjects had to concurrently track in these conditions was limited to three.

**High:** Each session involved a set of six images, all of which were encountered in both games. Consequently, in these conditions subjects had to concurrently track six images. To equalize low and high concurrent diversity conditions in terms of the number of images each image was pitted against within a game (two), as well as in terms of the total number of pairs of images between which subjects chose within each session, the six images formed only six different pairs. To enhance the impact of high concurrent diversity, unlike in the low concurrent diversity condition, images that were pitted against each other never had a common image that they were both pitted against (Table 1).

## Cumulative diversity

**Low**: Every image was pitted against the same two other images in two consecutive learning sessions. Thus, in total, subjects chose between each pair of images 32 times. In these conditions, subjects learned about twelve images in the span of two days.

**High**: Every image was pitted against two different pairs of images in two different learning sessions (i.e., against a total of four other images). Thus, over the same number of trials involving the same number of images, subjects encountered twice as many image pairs compared to the low cumulative diversity condition. Correspondingly, subjects chose between each pair of images 16 times, half as much as under low cumulative diversity. To ensure that the opportunity to learn relative preference was not hindered by a change in reward context midway through learning, reward context was always the same (i.e., either high or low) in both learning sessions of a given image. Implementing these criteria made it impossible to have subjects learn about twelve images in the span of two days, and thus, we had subjects learn about 18 images across the span of 3 days (Table 2). Furthermore, pairing each image with different images in two sessions meant that, for some stimuli, its two sessions could not be consecutive. Thus, 58.33% of high cumulative diversity were learned over the course of two days.

This extension of the learning period conferred a small benefit to accuracy in testing trials, as shown by a logistic regression on the number of days learning spanned (log odds accuracy improvement = .12 CI = [.05 .19]). Reassuringly, this incidental effect ran counter to the overall effect of high cumulative diversity, which was to impair testing performance. Thus, it did not change the interpretation of the main results.

**Table 1. Example image pairings in conditions of low and high concurrent diversity and reward context.**

**A) Concurrent diversity:** Low; **Reward context:** Low

| Stimulus | Expected value | Pitted against | Optimal choice frequency |
|---|---|---|---|
| 1 | 0 | 2 and 3 | Never |
| 2 | .33 | 1 and 3 | Half (over 1) |
| 3 | .66 | 2 and 3 | Always |

**B) Concurrent diversity:** Low; **Reward context:** High

| Stimulus | Expected value | Pitted against | Optimal choice frequency |
|---|---|---|---|
| 4 | .33 | 5 and 6 | Never |
| 5 | .66 | 4 and 6 | Half (over 4) |
| 6 | 1 | 4 and 5 | Always |

**C) Concurrent diversity:** High; **Reward context:** Low

| Stimulus | Expected value | Pitted against | Optimal choice frequency |
|---|---|---|---|
| 1 | 0 | 2 and 3 | Never |
| 2 | .33 | 1 and 4 | Half (over 1) |
| 3 | .66 | 1 and 5 | Always |
| 4 | .66 | 2 and 6 | Always |
| 5 | .33 | 3 and 6 | Half (over 6) |
| 6 | 0 | 4 and 5 | Never |

**D) Concurrent diversity:** High; **Reward context:** High

| Stimulus | Expected value | Pitted against | Optimal choice frequency |
|---|---|---|---|
| 7 | .33 | 8 and 9 | Never |
| 8 | .66 | 7 and 10 | Half (over 7) |
| 9 | 1 | 7 and 11 | Always |
| 10 | 1 | 8 and 12 | Always |
| 11 | .66 | 9 and 12 | Half (over 12) |
| 12 | .33 | 10 and 11 | Never |

To assess the formation of absolute values, we had subjects choose between images about which they had learned in two previous sessions ('testing' trials). Through the entire course of learning, such testing trials were interleaved with the learning trials (every 3rd trial, 24 testing trials total). Reward feedback was not shown on testing trials, but subjects were informed in advance that these trials would be rewarded with the same probabilities with which they were rewarded during learning. This reward was factored into the final bonus subjects received for their performance. Testing trials always presented a choice between images learned within the same condition. However, some of these images subjects had already chosen between during learning ('Learned Pair Trials'; 25% of testing trials), whereas other trials presented a choice between images about which subjects learned in separate games ('Novel Pair trials'; 75% of testing trials). Half of novel-pair trials were designed to assess how well subjects performed in general. These trials thus presented a choice between two images one which of was preferable to the other both in terms of reward probability and in terms of how it ranked in reward probability compared to the other images it was learned with. The other half of novel-pair trials were designed to distinguish between value and preference learning. Thus, half of these (25% of all novel-pair trials) presented a choice between images with the same expected value but that ranked differently in their original learning context, whereas the other half presented a choice between images with the same relative rank but different values. Pairs that satisfied these criteria were selected in random.

**Table 2. Example arrangement of images across days in conditions of high cumulative diversity.** Each number corresponds to an image subjects learned about. The arrangement ensured that every image would be pitted against four distinct images across two learning sessions Brackets group images that were pitted against each other. **A) Low concurrent.** Each game consisted of three images each pitted against the two other images. **B) High concurrent.** Each game consisted of six images, with each image pitted against two other images.

| A) Low concurrent | | |
| --- | --- | --- |
| **Day 1** | **Day 2** | **Day 3** |
| **Morning session** | | |
| **Game 1:** {1,2,3} | **Game 1:** {2,5,6} | **Game 1:** {14,15,18} |
| **Game 2:** {7,8,9} | **Game 2:** {8,11,12} | **Game 2:** {13,16,17} |
| **Evening session** | | |
| **Game 1:** {1,4,5} | **Game 1:** {3,4,6} | **Game 1:** {13,14,17} |
| **Game 2:** {7,10,11} | **Game 2:** {9,10,12} | **Game 2:** {15,16,18} |
| | | |
| **B) High concurrent** | | |
| **Day 1** | **Day 2** | **Day 3** |
| **Morning session (both games)** | | |
| {1,2},{1,3},{2,4}, {3,5},{4,6},{5,6} | {1,13},{1,14},{2,13} {2,15},{3,14},{3,15} | {4,14},{4,15},{5,13} {5,15},{6,14},{6,13} |
| **Evening session (both games)** | | |
| {6,7},{6,8},{7,9} {8,10},{9,11},{10,11} | {7,16},{7,17},{8,16} {8,18},{9,17},{9,18} | {10,16},{10,17},{11,17} {11,18},{12,16},{12,18} |

Within every session, we tested only the latest condition for which at least two learning sessions were completed. This meant that in most days only one condition was tested. However, equalizing the total number of testing trials across conditions required that there be on average 2 days (range 1–3) in which the morning and evening sessions tested different learning conditions. The variation between subjects emerged because of Sabbath observance, which resulted in some subjects completing only the morning session on Fridays (since sundown prevented the completion of the second session) and either subsequently continuing the following evening (Saturday evening) or the following day (Sunday morning).

In the morning session of the final day of the experiment (day 11), to ensure that there were sufficient testing trials of images learned on days 9 and 10, subjects were presented with testing trials from the last learned condition. In the afternoon session, subjects were presented with testing trials that spanned across conditions. However, not all subjects performed these afternoon trials, and some performed them incompletely. Therefore, the data from the afternoon trials were not included in the main analyses.

## Mobile platform

To test learning across multiple well-separated sessions, we modified an app developed by Eldar et al [32] for Android smartphones using the Android Studio programming environment (Google, Mountain View, CA). The app asks users to perform experimental tasks according to a predetermined schedule. Additional features of the app not relevant for the present work include probing of changes in subjects' mental state, including regular mood self-report questionnaires and life events and activities logging, and recording of electroencephalographic (EEG) and heart rate signals derived from wearable sensors connected using Bluetooth. All behavioral and physiological data are saved locally on the phone as SQLite databases (The SQLite Consortium), which are regularly uploaded via the phone's data connection to a dedicated cloud.

### Daily schedule

Subjects first visited the lab to receive instructions, test the app on their phones, and try out the experimental task (see Initial lab visit section below). Starting from the next day, subjects performed two experimental sessions a day, one in the morning and one in the evening, over a period of 10 consecutive days followed by a rest day, and a final day of testing. Each session began with a 5-minute heart rate measurement during which subjects were asked to remain seated. Following this, subjects put on the EEG sensor and played two games of the experimental task. The app allowed subjects to perform the morning session from either 6AM, 7AM, 8AM or 9AM, as best fitted the subject's daily schedule, and the evening session from 8 hours following this time. Subjects were allowed to adjust the timing of the sessions according to their daily schedule but were required to ensure a gap of at least 6 hours between successive sessions. On average, subjects performed the morning session at 8:56AM (mean SD ± 40 min) and the evening session at 6:12 PM (mean SD± 32 min). Subjects who were religiously observant were allowed to suspend the experiment due to holiday observance as long as they resumed it the following day. Twenty five out of twenty-seven subjects took a holiday break, but only six of those subjects took the break during learning about specific images (they had only completed one of two learning sessions with those images). We evaluated whether these breaks resulted in accuracy drop-off for these images relative to other images within the same condition for which learning was uninterrupted but found no significant drop-off ($Mean_{with\ break}$ = .87±.04, $Mean_{without\ break}$ = .84±.03; $p$ = .196 bootstrap test). As part of a larger data collection effort, subjects were also asked to report their mood prior to playing each game as well as twice more throughout the day.

### Materials

The experiment involved 60 images, which were abstract patterns collected from various internet sources. To ensure that images were sufficiently distinguishable from one another we ran a structural similarity analysis that assesses the visual impact of three characteristics of images: luminance, contrast, and structure [48]. We considered as sufficiently distinguishable images with a similarity index of at most .6, and this was verified by visual inspection.

### Statistical analyses

All non-modeling statistical analyses were performed in R using RStudio. Statistical tests were carried out using the bootstrap method with the "simpleboot" package. Correction for multiple comparisons across the two types of diversity was carried out using the Benjamini–Hochberg procedure.

### Regression analyses

Regression analyses were performed using the "brms" package, which performs approximate Bayesian inference using Hamiltonian Monte Carlo sampling. We used default priors and sampled two chains of 10000 samples each. 1000 samples per chain were used as warm-up. To ensure convergence, we required an effective sample size of at least 10000 and a R-hat statistic of at most 1.01 for all regression coefficients. To evaluate an effect of interest, we report the median of the posterior samples of the relevant regression coefficient and their 95% high density interval (HDI). A reliable relationship is said to exist between a predictor and an outcome if the 95% HDI excludes zero.

**Examining rank bias.**   To calculate whether subjects preferred images ranked higher in the original learning context, we calculated each subject's ranking of each image based on how many times they chose the image relative to the other images it was pitted against (best, second-best, or worst). If the difference in choice frequency between two images that were pitted against each

other was below 10%, indicating no established ranking between them, then the pair was excluded from the analysis (8% of trials). These rankings were then averaged across the two learning sessions in which an image was learned about to generate an overall rank for the image. Finally, we tested for a ranking bias by examining subjects' choice between differently ranked but similarly rewarded images (defined as less than a 10% difference in percent rewarded outcomes). Bias was defined as a tendency to choose the image that had been ranked higher during learning.

**Examining the influence of other options' outcomes.**    To determine whether training diversity modulated the influence on choice of past outcome of non-presently relevant images, we used a Bayesian logistic mixed model predicting subjects' choices. The predictors are described below for a choice between image A and image B:

1. *Own*—Reward history of the currently available images in the current pair, computed as the proportion of rewarded trials when option A was chosen versus any image minus the proportion of rewarded trials when option B was chosen versus any image.

2. *Current Alternative*—Reward history of the currently available images in the previous times the same images were pitted against each other, computed as the proportion of rewarded trials when option A was rejected in favor of option B minus the proportion of rewarded trials when option B was rejected in favor of option A.

3. *Other*—Reward history of choosing against the currently available images, computed as the proportion of rewarded trials when option A was rejected in favor of any image other than B minus the proportion of rewarded trials when option B was rejected in favor of any image other than A.

4. *Times Chosen*—Computed as the number of trials in which option A was selected out of all trials that included option A minus the number of trials when option B was selected out of all trials that included option B.

5. *Concurrent*—concurrent diversity condition.

6. *Cumulative*—cumulative diversity condition.

All two-way interactions between each type of reward history and each type of diversity condition: *Concurrent×Own, Concurrent×Current Alternative, Concurrent×Other, Cumulative×Own, Cumulative×Current Alternative, Cumulative×Other*.

To provide a concrete example, consider the following five-trial segment for which we calculate the corresponding value of each regressor:

Trial 1: A vs B, A is selected and rewarded

Trial 2: B vs C, B is selected and is not rewarded

Trial 3: B vs C, C is selected and rewarded

Trial 4: A vs C, A is selected and is not rewarded

Current trial. Trial 5: A vs C

"Own" is calculated as the proportion of times A was rewarded when A was chosen (Trial 1, Trial 4 for a total of A_own = 1/2) minus the proportion of times image C was rewarded when chosen (Trial 3; C_own = 1/1). Thus, $\beta_{own} = {}^1/_2 - 1 = -{}^1/_2$.

"Current alternative" is calculated as the proportion of times the subject was rewarded when they rejected image A in favor of image C (no such trial exists so A_current alternative is set to the null value of 1/2), minus the proportion of times the subject was rewarded when they rejected image C in favor of image A (trial 4 –C = 0). Thus $\beta_{current\ alternative} = 1$

"Other" is calculated as the proportion of times the subject was rewarded when they rejected image A in favor of any image other than C (no such trials exist so A_other = 1/2) minus the proportion of times the subject was rewarded when they rejected image C in favor of any image other than A. In our case, image C was rejected in favor of image B (trial 2) and image B is not rewarded (0/1) so the value of $\beta_{\text{other}} = \frac{1}{2}$

The probability of choosing image *A* over image *B* was thus modelled as:

$$
\begin{aligned}
\text{P(choice} = i) \sim \sigma(&\beta_0 + \beta_{\text{Own}}Own_{A-B}+ \\
&\beta_{\text{Current alternative}}Current\ Alternative_{A-B} + \beta_{\text{Other}}Other_{A-B}+ \\
&\beta_{\text{Times Chosen}}Times\ Chosen_{A-B} + \beta_{\text{Concurrent}}Concurrent+ \\
&\beta_{\text{Cumulative}}Cumulative + \beta_{\text{Concurrent}\times\text{Own}}Concurrent \times Own_{A-B} \\
&+\beta_{\text{Concurrent}\times\text{Current alternative}}Concurrent \times Current\ Alternative_{A-B}+ \\
&\beta_{\text{Concurrent}\times\text{Other}}Concurrent \times Other_{A-B} + \beta_{\text{Cumulative}\times\text{Own}}Cumulative \times Own_{A-B} \\
&+\beta_{\text{Cumulative}\times\text{Current alternative}}Cumulative \times Current\ Alternative_{A-B}+ \\
&\beta_{\text{Cumulative}\times\text{Other}}Cumulative \times Other_{A-B})
\end{aligned}
\tag{5}
$$

where $\sigma$ represents the logistic function. To account for between-subject variation, we included random intercepts as well as random slopes for all predictors.

## Computational formalization

Whereas the main components of the computational model are described in the main text, here we detail precisely how preference and value learning were influenced by diversity conditions. On each trial, a set of per-subject $\beta_{\text{preference–baseline}}$ and $\beta_{\text{value–baseline}}$ parameters were modulated by the following main effects and interaction parameters.

Main effects

$\beta_{\frac{\text{low}}{\text{high}}\text{concurrent}}$ and $\beta_{\frac{\text{low}}{\text{high}}\text{cumulative}}$ are ratios that represent the impact of concurrent and cumulative diversity during learning on choice.

$\beta_{\frac{\text{learning}}{\text{testing}}}$ is a ratio that represents how the influence of prior outcomes on choice differs between learning and testing trials.

Interaction terms

$\beta_{\frac{\text{low}}{\text{high}}\text{concurrent}\times\frac{\text{value}}{\text{preference}}}$ and $\beta_{\frac{\text{low}}{\text{high}}\text{cumulative}\times\frac{\text{value}}{\text{preference}}}$ are ratios that represent the impact of training diversity on the relative influence of value and preference learning on choice. The more either ratio diverges from 1, the greater the impact diversity has on the balance between value and preference learning.

$\beta_{\frac{\text{low}}{\text{high}}\text{concurrent}\times\frac{\text{learning}}{\text{testing}}}$ and $\beta_{\frac{\text{low}}{\text{high}}\text{cumulative}\times\frac{\text{learning}}{\text{testing}}}$ are ratios that represent the impact of training diversity on the relative influence of prior outcomes on choices in learning compared to testing trials; the more either ratio diverges from 1, the greater the impact diversity has on the relative influence of prior outcomes on choices in learning compared to testing trials.

$\beta_{\frac{\text{value}}{\text{preference}}\times\frac{\text{learning}}{\text{testing}}}$ is a ratio that represents the relative influence of value and preference learning on choice differs in learning and testing trials. The more this ratio diverges from one, the more preference and value learning are differentiated in the sense that one algorithm influences choices more during learning trials and the other algorithm influences choices more during testing trials.

Using these main effect and interaction parameters, $\beta_{\text{preference}}$ and $\beta_{\text{value}}$ can be computed for each trial type. Thus, for example, in a *learning* trial of *low concurrent* but *high cumulative*

*diversity* we can calculate the inverse temperature for the value and preference algorithm as follows:

$$\beta_{\text{preference}}=\beta_{\text{preference}-\text{baseline}} \times \beta_{\frac{\text{low}}{\text{high}}\text{concurrent}} \times \beta_{\frac{\text{learning}}{\text{testing}}}$$

$$\div \beta_{\frac{\text{low}}{\text{high}}\text{cumulative}} \times \beta_{\frac{\text{low}}{\text{high}}\text{cumulative} \times \frac{\text{value}}{\text{preference}}} \div \beta_{\frac{\text{value}}{\text{preference}} \times \frac{\text{learning}}{\text{testing}}}$$

$$\div \beta_{\frac{\text{low}}{\text{high}}\text{cumulative} \times \frac{\text{learning}}{\text{testing}}}$$

$$\div \beta_{\frac{\text{low}}{\text{high}}\text{concurrent} \times \frac{\text{value}}{\text{preference}}} \times \beta_{\frac{\text{low}}{\text{high}}\text{concurrent} \times \frac{\text{learning}}{\text{testing}}}$$

$$\beta_{\text{value}}=\beta_{\text{value}-\text{baseline}} \times \beta_{\frac{\text{low}}{\text{high}}\text{concurrent}} \div \beta_{\frac{\text{low}}{\text{high}}\text{cumulative}} \times \beta_{\frac{\text{learning}}{\text{testing}}} \div$$

$$\beta_{\frac{\text{learning}}{\text{testing}} \times \frac{\text{low}}{\text{high}}\text{cumulative}} \div \beta_{\frac{\text{low}}{\text{high}}\text{cumulative} \times \frac{\text{value}}{\text{preference}}} \times \beta_{\frac{\text{learning}}{\text{testing}}}\beta_{\frac{\text{low}}{\text{high}}\text{concurrent}} \times$$

$$\beta_{\frac{\text{low}}{\text{high}}\text{concurrent} \times \frac{\text{value}}{\text{preference}}} \times \beta_{\frac{\text{value}}{\text{preference}} \times \frac{\text{learning}}{\text{testing}}}$$

(6)

## Alternative models

To identify the computations that guided subjects' choices we compared the model presented in the main text to several variations of this model, in terms of how well each fitted subjects' choices. These included a model that only learns absolute values ($\beta_{\text{preference}} = 0$), a model that only learns relative preferences ($\beta_{\text{value}} = 0$), a model with leak parameters ($\gamma_{\text{preference}}$ and $\gamma_{\text{value}}$) that vary across conditions (BIC +240), and a non-Bayesian learning model that, instead of beta distributions, learns expected values and relative preferences based on a Rescorla-Wagner update rule with fixed learning rates. This latter model includes all main-effect and interaction parameters included in our winning model except for the learning rates $\alpha_{\text{value}}$, $\alpha_{\text{preference}}$, which replaced the leak parameters $\gamma_{\text{value}}$, $\gamma_{\text{preference}}$ ([49]; BIC +2370 relative to winning model).

To test for a gradual shift towards absolute-value encoding as a function of time (see Ruling Out Alternative Explanations), we implemented a model that scales the inverse temperature parameter for value learning, $\beta_{\text{value}}$, by $e^{\tau \cdot \text{trial}(i,t)}$, and the inverse temperature parameter for preference learning, $\beta_{\text{Preference}}$, by $\frac{1}{e^{\tau * \text{trial}(i,t)}}$. Here $\tau$ is a free parameter controlling the degree of the shift, as a function of the number of trials that elapsed since image i was chosen at trial t. These inverse temperatures apply specifically to the outcome obtained that trial.

To additionally determine whether our data might be better accounted for by prior work which suggested that humans gradually shift towards value learning in similar RL experiments [39], we implemented an alternative model which assumes that the shift towards absolute-value encoding grows with the number of trials that elapsed since the beginning of the experiment. This model, too, did not fit the data well as the original model that did not assume a continuous gradual shift (+BIC 353), likely because subjects were made aware from the beginning of the experiment that they would need to generalize learned values.

Finally, we tested a beta-binomial model which allowed for asymmetries in the learning process from rewarded and non-rewarded outcomes. This model indeed improved the fit to the data but did not alter any of the main findings (S3 Fig)

## Model fitting

We fit model parameters to subjects' choices using an iterative hierarchical importance sampling approach [32] using MATLAB. We first used $2.5 \times 10^5$ random settings of the parameter

from predefined group-level distributions to compute the likelihood of observing subjects' choices given each setting. We approximated posterior estimates of the group-level prior distributions for each of our parameters by resampling the parameter values with likelihoods as weights, and then re-fit the data based on the updated priors. These steps were repeated iteratively until model evidence ceased to increase. To derive the best-fitting parameters for each individual subject, we computed a weighted mean of the final batch of parameter settings, in which each setting was weighted by the likelihood it assigned to the individual subject's decision.

## Parameter initialization

Across both models, $\beta_{\text{preference–baseline}}$ and $\beta_{\text{value–baseline}}$ were initialized by sampling from a gamma distribution ($k = 1$, $\theta = 1$), leak parameters ($\gamma_{\text{preference}}$ and $\gamma_{\text{value}}$) were initialized by sampling from a beta distribution with ($\alpha = 9$ $and$ $\beta = 1$) and all other parameters were initialized by sampling from a lognormal distribution ($\mu = 0$, $\sigma = 1$).

## Model comparison

For each model we estimated the optimal parameters by likelihood maximization. We then applied the Bayesian Information Criterion (BIC) to compare the goodness of fit and parsimony of each model. A so-called 'integrative BIC' [50] can be computed as follows: BIC = - 2 ln L + k ln n, where L is the evidence in favor of each model, estimated as the mean likelihood of the model given random parameter settings drawn from the fitted group-level priors, k is the number of fitted group-level parameters and n is the number of subject choices used to compute the likelihood. This method has shown high reliability and efficacy in detecting differences within and between subjects [50–53]. We validated the model comparison procedure by simulating data using each model and using the model comparison procedure to recover the correct model (S3 Table). To validate the BIC model comparison results, we also performed model comparison using the Akaike information criterion (AIC) (S5 Table).

## Statistical tests of parameter fits

Statistical significance of each interaction parameter was measured using a two tailed permutation test. First, we calculated the mean of the log fit across the 27 subjects to generate a summary statistic of how much the parameter deviates from 1. A mean of zero indicates that the parameter of interest does not significantly scale the inverse temperature in either direction. A mean significantly different from zero indicates the condition modulates the inverse temperature parameters in favor of either absolute values or relative preferences.

Thus, for each parameter of interest, we generated a null distribution composed of 1000 random permutations of the data, randomly shuffling the condition of interest (e.g., whether the subject is in low or high concurrent diversity, which corresponds to inverting the impact of the parameter). We then applied the full model fitting procedure to each permuted data set and computed the p value by comparing the actual parameter fit to the distribution of parameter fits for the permuted data. We validated our parameter fits through simulating data using the best fitting parameters for each subject and then recovering those parameters (S4 Table).

## Falsification of alternative models

In addition to model comparison, we examined whether alternative models generate predictions that are falsifiable by the data. Simulations from a Preference-only model could not account for subjects' ability to perform generalization at a high level across conditions

(Posterior model prediction: mean = .66 ±.1 vs Real data mean = .83±.2.), whereas a value-only model could not account for the difference in accuracy between learned-pair and novel-pair trials

(Mean$\Delta_{\text{concurrent low}}$ = 0±1, Mean$\Delta_{\text{concurrent high}}$ = 0±.1; p = .55 bootstrap). Furthermore, such a model was unable to account for the effects of other images' outcomes on image choices. Specifically, the effect of other image's outcomes was not significant CI = [-5.31,4.84] nor was it modulated by concurrent diversity CI = [-4.45,5.76]. Thus, neither value learning nor preference learning could account for subjects' behavior alone.

## Dryad DOI

https://doi.org/10.5061/dryad.1rn8pk0xr [46]

## Supporting information

**S1 Table. Performance on testing trials.**
(DOCX)

**S2 Table. Bayesian logistic regression of subjects' choices on reward history, training diversity, and past choice.**
(DOCX)

**S3 Table. Model validation.** 10 full experimental datasets were simulated using each model. Rows indicate the model used to simulate data and columns indicate the model recovered from the data using the model comparison procedure.
(DOCX)

**S4 Table. Validation of Parameter Recovery.** We validated our parameter fits through simulating data using the best fitting parameters for each subject and then recovering those parameters. Our correlation between simulated and recovered parameters was at least .74 for all parametes of interest that capture the effects of the experimental conditions, and at least .51 for all other parameters.
(DOCX)

**S5 Table. Model comparison using AIC.** To validate the BIC model comparison results, we also performed model comparison using the Akaike information criterion (AIC).
(DOCX)

**S1 Fig. Choice accuracy on similarly ranked images with different expected values across diversity conditions.** Concurrent diversity improved subjects' performance (Mean_high = .83 ±.02; Mean_low = .78±.02; p$_{\text{corrected}}$ = .035 bootstrap test). By contrast, cumulative diversity impaired performance on such trials (Mean_low = .82±.02 Mean_high = .77±.02 p$_{\text{corrected}}$ = .04 bootstrap test), as consistent with its general effect on overall performance. The plot shows individual subject accuracy (circles), group distributions of accuracy levels (violin), group means (thick lines) and standard errors (gray shading).
(TIF)

**S2 Fig. Data simulated using the combined value and preference learning model demonstrates all key behavioral findings.** To determine whether the model successfully captured individual differences in our experiment, we examined how parameter fits correlated with model-agnostic measures of behavior. As expected, we found that $\beta_{\text{value}}$ was significantly correlated with generalization performance (r = .7) while $\beta_{\text{preference}}$ correlated with our measure of rank bias (r = .5). We then validated the best-fitting model thoroughly by simulating, for

each subject, 1000 data sets using their best fitting parameters and analyzing the simulated data in the same fashion in which we analyzed the real data. This procedure showed the model uniquely accounted for all of our behavioral findings (Fig 1) (A) In learning trials, performance is better in conditions of low concurrent (Mean$_{low}$ = 88% ±1% vs Mean$_{high}$ = 85%±1) and cumulative (Mean$_{low}$ = 85%±1% vs Mean$_{high}$ = 82%±1%) diversity. (B) Concurrent but not cumulative diversity leads to better generalization (Concurrent: Mean$_{low}$ = −7.8%±1% vs Mean$_{high}$ = −1.2%±1 p$_{corrected}$ < .001 bootstrap test; Cumulative Mean$_{low}$ = −4%±1% vs Mean$_{high}$ = −6%±1 p$_{corrected}$ = .23). (C) Concurrent but not cumulative diversity diminishes ranking bias (Concurrent: Mean$_{low}$ = 59% ±1% vs Mean$_{high}$ = 53%±1, p = .01, Cumulative: Mean$_{low}$ = 56% ±1% vs Mean$_{high}$ = 54%±1). (D) The simulated choices show that preference for an image is inversely influenced by the outcomes of both the current alternative ($\beta_{Current\ alternative}$ = -.46, CI = [-.56, -.35]) and of the other images it had previously been pitted against ($\beta_{Other}$ = -.42, CI = [-.54, -.30]). Furthermore, the influence of other images' reward history is reduced by high concurrent diversity ($\beta_{concurrent \times other}$ = .31, CI = [.22, .40]).
(TIF)

**S3 Fig. Model fits of the winning model with asymmetric learning rates.** To account for previous findings of asymmetric in learning from positive versus negative reward prediction errors [54] we implemented a beta binomial model with asymmetric update rates. This modification did not alter any of the main findings. Namely, preference learning manifested to a greater extent in conditions of low concurrent diversity (low concurrent: $\beta_{preference}$ = 3.07 ±.21; high concurrent: $\beta_{preference}$ = 1.76 ±.12; *p*<.001 permutation test) whereas value learning manifested to a greater extent in conditions of high concurrent diversity (low concurrent: $\beta_{value}$ = 4.04±.18; high concurrent: $\beta_{value}$ = 4.54±.14 *p*<.001 permutation test). Furthermore, as in our winning model, cumulative diversity inhibited value learning (low cumulative: = 4.99 ±.3; high cumulative: $\beta_{value}$ = 3.78 ±.3; p < .001 permutation test) and had no significant impact on preference learning (low cumulative: $\beta_{preference}$ = 1.60±.12; high cumulative: $\beta_{preference}$ = 1.87±.14; p = .11 permutation test)
(TIF)

## Acknowledgments

We thank Elisa Milwer and Alina Ryabtev for help with the data collection process.

## Author Contributions

**Conceptualization:** Levi Solomyak, Paul B. Sharp, Eran Eldar.

**Data curation:** Levi Solomyak.

**Formal analysis:** Levi Solomyak, Eran Eldar.

**Resources:** Levi Solomyak, Paul B. Sharp, Eran Eldar.

**Supervision:** Eran Eldar.

**Writing – original draft:** Levi Solomyak, Paul B. Sharp.

**Writing – review & editing:** Levi Solomyak, Paul B. Sharp, Eran Eldar.

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
