## [Decision Letter · Decision Letter 0]

22 Jul 2022

Dear Mr. Solomyak,

Thank you very much for submitting your manuscript "Training diversity promotes absolute-value-guided choice" for consideration at PLOS Computational Biology.

As with all papers reviewed by the journal, your manuscript was reviewed by members of the editorial board and by four independent reviewers.  In light of the reviews (below this email), we would like to invite the resubmission of a significantly-revised version that takes into account the reviewers' comments. We cannot make any decision about publication until we have seen the revised manuscript and your response to the reviewers' comments. Your revised manuscript is also likely to be sent to reviewers for further evaluation.

Sincerely,

Lusha Zhu, Ph.D.

Associate Editor

PLOS Computational Biology

Samuel Gershman

Deputy Editor

PLOS Computational Biology

Reviewer's Responses to Questions

**Comments to the Authors:**

Reviewer #1: Thank you for the opportunity to review this manuscript by Solomyak and colleagues, which reports the results of an interesting longitudinal experiment studying the factors that influence absolute-value-guided choice versus relative-value-guided choice. This topic speaks to a recent debate in the reinforcement learning literature, which distinguishes between (absolute-)value-based learning models, in which participants are assumed to learn a representation of the discounted cumulative expected reward associated with different options, versus relative-value (or preference-based) learning models, in which participants learn a representation of the relative goodness of different choice options, but not their absolute value. This manuscript argues that one situational factor that promotes learning of absolute values is 'concurrent diversity', operationalized as the number of different alternatives that a choice option is paired with during the learning phase.

Strengths of the paper include its careful longitudinal research design and its use of sophisticated modelling analyses to get at the aspects of the data that address its research question. I have several questions about the manuscript that are mostly about ascertaining that the statistical analyses do indeed provide an appropriate degree of support for the manuscript's conclusions.

- One of the primary model-agnostic statistics in support of the manuscript's conclusions (reported on Page 14, lines 176-182) is that at the test phase, participants preferred similarly rewarding higher-ranked images from low-concurrent-diversity conditions (Mean = 63%, p = .006) but not from the high-concurrent-diversity condition (Mean = 49%, p = .24). First of all, these numbers should be tested against one another to ensure that this difference is statistically significant. Secondly, my impression is that this test does not account for the actual reward rates of the different stimuli (i.e., the higher-ranked image might have a reward rate of 65% and the lower-ranked image might have a reward rate of 56%). It is therefore crucial to control for the actual reward rate differences of the two stimuli in the test phase in this analysis, and to show that the difference between these two conditions remains statistically significant once the actual reward rate differences of the two stimuli are accounted for.

- The beta-binomial model implicitly assumes that participants learn equivalently from gains and losses (Equation 1 and Equation 2). Other beta-binomial models are possible which allow for asymmetries in the learning process (e.g., the count of reward outcomes may be incremented at a different rate to the count of non-reward outcomes). Given the numerous studies showing asymmetries in learning from positive versus negative reward prediction errors (including in a transfer learning context; see Ciranka et al., Nature Human Behaviour, 2022), it could be useful to include such an asymmetry in the learning models.

- In the 'Testing Alternative Interpretations' section on Page 21, the manuscript tests whether there is a shift in value learning over the course of learning by means of a variant model in which there is a difference in the absolute value learning parameter only in the second half of training. This stepwise change model seems to me to be not giving a fair shot to this alternative explanation - what if the change is gradual, rather than occurring all of a sudden half way through the task? I suggest that a stronger test of this alternative explanation would be if the beta_preference and beta_value parameters were permitted to vary linearly across the course of training (with an intercept and slope fit to the data) rather than assumed to be constant or varying in a stepwise fashion.

Reviewer #2: Solomyak and colleagues investigated the environmental conditions that promote the learning of absolute value or relative preferences among choice options, both of which were common strategies in humans’ decision-making. The article investigated four learning environments and tested the participants’ performance in both learnt and novel pairs. Interestingly, the article found that training diversity promoted absolute-value-guided choice. However, a few major issues listed here needed to be addressed.

A major concern was whether the results of the current study could be explained by a third hypothesis: ranked preference. Specifically, the participants might learn the relative preference but also make use of the transitivity of relative preferences. That is, participants might learn the rank of items, but not necessarily the absolute value of each item. For example, if participants learnt A was better than B and B was better than C in the training phases, then participants would easily infer A was better than C, even in situations where the participants did not learn the value of A and C directly. Whether there is evidence supporting the absolute value strategies that cannot be explained by this hypothesis?

A second concern was the contributing factor to the generalization effect. The article found the generalization effect by calculating the accuracy differences between the novel pairs and the learnt pairs (Figure 3). It was not clear whether the generalization effect was caused by the higher performance in the novel pairs or lower performance in the learnt pairs. It might be helpful to do further analysis to confirm the contributing factor to the generalization effect and report the effect size in all four conditions to exclude the possibilities that only some of the conditions contribute the effect.

Minor Concerns:

It is not clear whether statistical tests in this article had been corrected for multiple comparisons (e.g., The test in Figure 3 needed to be corrected, since the same data was used to test the two hypotheses, concurrent diversity and cumulative diversity).

There are some typos in the manuscript: (i) In the caption of Figure 2, the t value was missing, “t(51.8)”; (ii) Figure citations need to be consistent (e.g., Line 113, “Figure 2b” to Figure 2B); (iii) Figure 4 was not cited in the main text.

Some preprint reference has been published and might be updated. (e.g., “Hayden B, Niv Y: The case against economic values in the brain. PsyArXiv 2020”, now is Hayden BY, Niv Y. The case against economic values in the orbitofrontal cortex (or anywhere else in the brain). Behavioral Neuroscience. 2021;135: 192–201. doi:10.1037/bne0000448)

Reviewer #3: The paper poses an excellent question and will be a good addition to the literature once the authors expand and clarify the methods. The analyses also need to be stregthnend. At the moment, it is unfortunately difficult to ascertain the particulars the experimental procedure, which somewhat reduces the interpretability of the results. This is particularly worrying as we (this review is a joint work) work in the field of relative/absolute value learning.

Please see our detailed comments below:

Major comments:

Task Description

Options/conditions

1/ It is not very clear what options participants saw when and what was their visual representation. It would be helpful to have a figure similar to Figure 1C showing the option combinations for each day, condition, session and game, explicitly stating both the outcome probability of each option as well as the image (or image id) shown to the participant. Also note that Figure 1B&C suggests a different set-up of your task than I believe was the case. Namely, it suggests that in the low concurrent condition, games in the same session used the same options (and images) and that options/images were repeated across the conditions.

Experimental testing

2/ Very little is said about the particulars of the testing trials. Can you please include more details? Here are some of the questions we had:

What was the ratio of learnt vs novel pairs?

How many repeats were there per pair?

Was day 11, the only day you tested images from multiple conditions or was this also the case on previous days?

Were the paired options originally from the same day/condition or did you combine options across day/conditions?

How did you come up with the novel pairs? Did you have any criteria when drawing the novel pairs or were the pairings random, or exhaustive (all possible combinations)?

While feedback was not shown for the testing trials, were the testing trials rewarded? And if so how?

Also there is some information that you mention in the figure legend but not in the methods themselves. Could you please put all relevant info in the methods even if it means repeating it? E.g. Figure 1C - There were 24 testing trials per game

Data analysis

3/ Additional Analysis required

It would be great if you could split novel pairs testing into groups based on the relative and absolute value prediction and show the bias separately for each group. In particular the following grouping will be of particular interest:

- A group with options that have the same absolute value but differ in the relative value (I believe you already isolated those and plotted them in Figure 4A?)

- A group with pairs where the relative value is the same (or very similar) for both options but their absolute value is different, e.g. options like 100% vs 66% from low reward context and 66% from high reward context vs 33% from low reward context

- A group where one of the options has both higher relative and absolute value, e.g. 100% vs 33% low reward context. If I understood your set-up correctly, this should be the majority of novel pairs.

Potentially, also a group where the option with the higher absolute value has lower relative value (though I don’t think there was any pair like this in your task. The reason for this is that while all 4 groups are informative of participants ability to generalise, only those in group 1,2 & 4 are actually informative of their preference for absolute or relative encoding.

4/ Did you observe any difference in responses to the testing trials across the span of the experiment? Did participants become more absolute encoders over time as they learnt that they will be tested on novel pairs (as they did in Juechems 2021)

Sanity checks:

5/ In choice sets where the 100% option was present, how often participants explored the alternative option.

6/ What about the simulations the alternative models? Are they falsified by the data?

Missing details

7/ (lines 182 - 184) to prove your point, you should test that preference for higher ranked images in the low concurrent condition was significantly higher than in the high concurrent condition (and the same for the cumulative condition). If one group is significantly different from a reference value while the other group is not, it does not mean that the group difference is significantly different

8/ (lines 147 - 154). How did you test this?

Previous work and position of the paper within the literature.

9/ The authours should refer to Haynes and Wedell (“in and out of context”; Journal of Experimental psychology).

10/ The authors should mentioned within lines 51-52 “authors have proposed models of” a reference to Palminteri and Lebreton 2021, where most of the relative value learning models are summarized.

11/ Even more importantly, the proposed winning model is a variant of the model proposed by Bavard et al. (2018; Nature Communications; HYBRID), however somehow surprisingly this is not acknoledged and the paper is not cited.

12/ Bavard et ak. (2021; Science Advances) propose a model to explain why context effects (i.e., deviation from absolute values) are stronger in blocked concerning interleaved design (the idea is global versus local context values). Is this idea relevant here?

Minor comments:

- The fraction of your code that is publicly available is written in MATLAB as opposed to R you mentioned in the paper. - - Did you do part of modelling in MATLAB? If so, can you specify what optimizer (and other functions) you used?

- The distinction you make between concurrent and cumulative conditions was not very clear from the introduction. Although, Figure 1 somewhat clarified the issue, rephrasing the relevant lines in the introduction (72-75) would be beneficial, alternatively adding an example might also help.

- Figure 1A - there is no ITI in the picture, I believe that a game had 48 trials instead of the 72 mentioned in the figure.

- 109-111 - This is calculated on data from the learning trials right? I.e. trials with feedback

- 129 - not sure I understand what a proportion reward difference is in this case. Can you please elaborate?

- 179 - p=.006 is missing a period before the 00

- 184 - If you stated in the previous section that cumulative condition had no effect on absolute vs relative learning. Why is it suddenly problematic that you do not find any effect of the cumulative condition on a related measure .

- 268 - Is the βvalueV(i) + βpreferenceW’(i) ever over 1?

- 290-291 - β preference is stated twice with different results. I assume one of them was supposed to be β value?

- 279 - Did you fit the model only on the learning data or also on the testing data?

- 327- Did you get the same results with AIC?

- Supplementary Table 2 - Low Concurrent - Day 2 - Game 1 - Wasn't this supposed to be {4,5,6} and {10,11,12}? as per the previous schedule?

- 539 - Did you observe differences in performance in the test trials during the regular 10-day schedule and during day 11 (i.e. after one day break?)

- 549 - How many people took the break for holiday observance, did it fall within or between condition and did it affect the performance in any way?

- 878 - Can you include the model you used to calculate the rank bias?

- 635 - choice difference?

- 644 - Could you make the gap between the first fraction and the second fraction larger? On a first read it looks like a single fraction.

- 653 - Can you include a full description of the RW model you used, including free parameters?

Did you also use R for the model fitting. IF you which package?

Reviewer #4: Solomyak et al present an interesting and ambitious experiment testing how two different sources of diversity, concurrent and cumulative, push participants to learn the expected or relative value of a choice over 10 days. They found that concurrent, but not cumulative, diversity helps participants encode the expected versus relative value of an option by enabling better generalization when making new choices between previously learned options. Nevertheless, this manuscript could substantially benefit from major revisions. Namely, it is challenging to adequately assess the results and their interpretation when the task details are unclear.

Major points

1. I appreciate that this is a complicated task to describe, but it is currently very difficult to understand task details and conditions, and in some cases, task visualization appears to conflict with task description:

— For example, (line 482) in the low concurrent diversity condition, two games within each learning sessions are said to each have a distinct set of three images, yet in the experimental design figure (figure 1), it appears that they share the same three images. While I understand that the figure may not describe exact task conditions, it is extremely confusing and should mirror the actual task as much as possible. From the figure, one would infer that some options (i.e., orange and green) are sampled every day, across conditions and much more frequently than other options (i.e., hot pink and light green). From the methods, it appears that there were 60 stimuli, did participants learn all 60? How many times did they get to see each stimulus? Over how many days was each stimulus presented? I know the latter differs on whether cumulative diversity was high or low, but even in the table example on page 32, it appears that some stimuli are repeated across days, and some are not - how was this determined? These task details need to be made much clearer and in the main manuscript (the experimental design figure should also include this information).

— (line 508) It is mentioned that for high cumulative diversity, the reward context was the same in both learning sessions of a given image. Was this true of other conditions? What was the high/low reward context breakdown by condition/day? (could also be noted in experimental design).

2. From what I understand, testing either occurred interleaved during learning or during the last testing day. How were the novel pairs chosen? It is mentioned that the stimuli were paired with stimuli from prior sessions (‘two sessions’ back), but was this restricted to the same condition? Additionally, how were the novel pairs chosen on the last day? I see they were drawn from different conditions, did this lead to different choice behavior than during interleaved testing? How did older versus newer options fare on Day 11 testing? (i.e., how did long-term memory influence choice?).

3. To rule out the potential confound that a difference in the duration of learning is modulating the high versus low concurrent condition effects, authors tested a model where the shift towards absolute value learning only occurred in the second half of high-concurrent sessions. Since this model fit worse, they ruled out the confound. While I agree with this approach and interpretation, it belies an interesting assumption that more learning should lead to more ‘absolute’ versus relative preference. From what I can glean from the manuscript, the full model doesn’t seem to incorporate a dynamic “shift” from relative to absolute values, right? I say this because of the large literature on habits showing that, conversely, it could be expected that more learning leads to stronger habits/relative preference. Perhaps the potential/speculated dynamic between expected and relative value learning could be expanded upon in the discussion. Alternatively, if there is a way to test a shift from relative to absolute value learning in the current data, that should be examined.

4. A fascinating tension in the concurrent condition is that while (as expected) learning is worse in the high condition, generalization is better. Is this observed at an individual-difference level? Are worse learners better able to generalize? It could be an interesting consequence of this trade-off.

5. To differentiate independent versus relative value and given the literature on learning ‘extreme’ values within a learning context, were there test trials between the ‘extreme’ options in the two reward contexts? i.e., comparing choice between (1) stimulus associated with 0 reward (low-reward context) and stimulus associated with 0.33 probability reward in the high-reward context and (2) stimulus associated with 0.66 reward probability in low-reward context versus the 100% probability reward stimulus (high-reward context). If so, is there a reliable difference in choice for these pairs in the different conditions?

Minor points

1. Equations 6 and 7 are difficult to read. For equation 6, you may want to have consistent notation with the paragraph above (a/b versus i/j), and I can’t parse equation 7, there seem to be missing/inconsistent notation of operations.

2. It appears that only some of the effects of the Bayesian logistic mixed model (equation 6) are reported, it would be helpful to have a full table or report of the results.

3. The description of “own” versus “current alternative” and “other” isn’t clear or very intuitive. I think it would help to have a clear example of the three calculations and their distinctions.

4. The language describing “absolute” values could be slightly misleading since “absolute” is often used to mean unsigned (i.e., a value of 5 and -5 both have an absolute value of 5). I don’t think you need to change this language throughout the paper, but comparing “expected (or ‘independent’) value” and “relative value” may be more straightforward.

5. (line 82) It is first said that consecutive sessions were separated by 12 hours on average, but later (line 547) it appears this difference was (at least 6) and closer to 9 hours, and the average hours are also different in the experimental design figure description (6:56 versus 6:12).

6. I think there is a typo in Table 1 C) for stimulus 5 - it says it’s pitted against 3 and 7, but stimulus 7 isn’t included in that set.

7. (line 65) “Unfamiliar sets of familiar options” may be better understood as “New or novel sets” of learned or experienced options.

**Have the authors made all data and (if applicable) computational code underlying the findings in their manuscript fully available?**

Reviewer #1: Yes

Reviewer #2: None

Reviewer #3: **No: **there is discrepancy between what they say in the manuscript (R) and what is shared (Matlab) that has to be clarified.

Reviewer #4: None

PLOS authors have the option to publish the peer review history of their article (what does this mean?). If published, this will include your full peer review and any attached files.

Reviewer #1: No

Reviewer #2: No

Reviewer #3: No

Reviewer #4: No
---

## [Decision Letter · Decision Letter 1]

18 Oct 2022

Dear Mr. Solomyak,

We are pleased to inform you that your manuscript 'Training diversity promotes absolute-value-guided choice' has been provisionally accepted for publication in PLOS Computational Biology.

Best regards,

Lusha Zhu, Ph.D.

Academic Editor

PLOS Computational Biology

Samuel Gershman

Section Editor

PLOS Computational Biology

Reviewer's Responses to Questions

**Comments to the Authors:**

Reviewer #1: I thank the authors for their thoughtful response, which has addressed my concerns regarding this manuscript.

Reviewer #2: The authors addressed all of my concerns. I have no further issues and appreciate that the authors performed additional analyses to convey their points well.

Reviewer #3: After careful consideration, I believe the authors successfully addressed my concerns.

Reviewer #4: The authors provided a clear and thorough response to my comments, no further suggestions.

**Have the authors made all data and (if applicable) computational code underlying the findings in their manuscript fully available?**

Reviewer #1: Yes

Reviewer #2: Yes

Reviewer #3: **No: **I could not find an active link

Reviewer #4: Yes

PLOS authors have the option to publish the peer review history of their article (what does this mean?). If published, this will include your full peer review and any attached files.

Reviewer #1: No

Reviewer #2: No

Reviewer #3: No

Reviewer #4: No

---

## [Editor Report · Acceptance letter]

25 Oct 2022

PCOMPBIOL-D-22-00824R1 

Training diversity promotes absolute-value-guided choice

Dear Dr Solomyak,

I am pleased to inform you that your manuscript has been formally accepted for publication in PLOS Computational Biology. Your manuscript is now with our production department and you will be notified of the publication date in due course.

With kind regards,

Anita Estes
